# Association between antenatal care utilization pattern and timely initiation of postnatal care checkup: Analysis of 2016 Ethiopian Demographic and Health Survey

**Gizachew Tadesse Wassie**[1]*, **Minyichil Birhanu Belete**[2], **Azimeraw Arega Tesfu**[3], **Simachew Animen Bantie**[3], **Asteray Assmie Ayenew**[3], **Belaynew Adugna Endeshaw**[4], **Semaw Minale Agdie**[5], **Mengistu Desalegn Kiros**[6], **Zelalem T. Haile**[7], **Mohammad Rifat Haider**[8], **Gillian H. Ice**[9,10]

1 Department of Epidemiology and Biostatistics, School of Public Health, College of Medicine and Health Science, Bahir Dar University, Bahir Dar, Ethiopia, 2 Department of Pediatrics and Child Health Nursing, School of Health Sciences, College of Medicine and Health Science, Bahir Dar University, Bahir Dar, Ethiopia, 3 Department of Midwifery, School of Health Sciences, College of Medicine and Health Science, Bahir Dar University, Bahir Dar, Ethiopia, 4 Department of Physiotherapy, School of Medicine, College of Medicine and Health Science, Bahir Dar University, Bahir Dar, Ethiopia, 5 Department of Gynecology and Obstetrics, School of Medicine, College of Medicine and Health Science, Bahir Dar University, Bahir Dar, Ethiopia, 6 Department of Anatomy, School of Health Sciences, College of Medicine and Health Science, Bahir Dar University, Bahir Dar, Ethiopia, 7 Department of Social Medicine, Heritage College of Osteopathic Medicine, Ohio University, Dublin, Ohio, United States of America, 8 Department of Social and Public Health, College of Health Sciences and Professions, Ohio University, Athens, Ohio, United States of America, 9 Department of Social Medicine, Heritage College of Osteopathic Medicine, Ohio University, Athens, Ohio, United States of America, 10 Global Health Initiative, College of Health Sciences and Professions, Ohio University, Athens, Ohio, United States of America

* leulgzat@gmail.com

**Data Availability Statement:** All relevant data are within the paper.

## Abstract

### Background

Despite significant public health intervention, maternal mortality remains high in low- and middle-income countries, including Ethiopia. Effective postnatal care is a critical service to reduce maternal mortality. In Ethiopia, only 17% of mothers received postnatal care services in 2016.

### Objective

This study examined the association between antenatal care and timely postnatal care checkup among reproductive-age women in Ethiopia.

### Methods

The study used the 2016 Ethiopian Demographic and Health Survey data. The current study included 4,081 women who give birth in the two years preceding the survey. Chi-square test and multivariable logistic regression analyses were used to examine the association between antenatal care and timely initiation of postnatal care.

**Funding:** The author(s) received no specific funding for this work.

**Competing interests:** The authors have declared that no competing interests exist.

## Results

Postnatal care services within 2 days of delivery were received by 16.5% of women. Women who had at least four timely antenatal care visits had higher odds of timely postnatal checkups compared to women who had no antenatal care [adjusted Odds Ratio (aOR): 2.50; 95% CI 1.42–4.42]. Women who had at least four antenatal care visits without timely initiation also had higher odds of postnatal check-up than their counterparts (aOR: 2.46; 95%CI: 1.22–4.97). Other factors significantly associated with timely initiation of PNC were secondary and above education (aOR: 1.64; 95%CI: 1.03–2.60), perceived distance to the nearby health facility as a significant barrier (aOR: 1.55; 95%CI: 1.15–2.09), primiparous (aOR: 0.34; 95%CI: 0.19–0.61) and institutional delivery (aOR: 14.55; 95%CI: 2.21–95.77).

## Conclusion

The prevalence of timely initiation of postnatal care in Ethiopia is very low. Women who received recommended antenatal care services had higher odds of timely initiation of post-natal care. Thus, strengthening the existing maternal and child health programs to adhere to the recommended ANC care guidelines may improve the timely initiation of postnatal care.

## Introduction

The postnatal period, defined as the first six weeks after birth, is critical to the health and survival of mother and newborn [1]. The majority of maternal deaths, mainly due to hemorrhage, and many newborn deaths, mostly due to asphyxia, occur during the first day of birth [2, 3]. Other complications following childbirth that may affect women in the postpartum stage including sepsis, cardiovascular events, deep vein thrombosis, stroke, and embolism require immediate medical attention [4]. Furthermore, lack of care during this period may result in disability as well as create missed opportunities for promoting healthy behaviors affecting women, newborns and children [1, 5].

Although global maternal deaths have decreased by 43% from 532,000 in 1990 to 303,000 in 2015 [6], maternal morbidity and mortality remain a major challenge to health care systems. Worldwide, the maternal mortality rate is 216 per 100,000 live births with a large proportion of deaths due to early preventable or treatable pregnancy and childbirth complications. Low- and middle-income countries (LMIC) account for 99% of maternal deaths, with sub-Saharan African countries alone accounting for 66% [7, 8]. In Ethiopia, in 2016 the maternal mortality ratio (MMR) remains high which accounts for 412 maternal deaths per 100,000 live births [9].

In the continuum of maternal health care, antenatal care (ANC) and postnatal care (PNC) are the key strategies required to reduce maternal and newborn deaths [10]. ANC is an entry point for maternal and child care service utilization, and as such, has the capability of reducing both maternal and neonatal mortality by detecting at-risk pregnancy and managing the risk associated. ANC provides an opportunity to adequately prepare mothers for birth and for appropriate care of children by addressing relevant information and education concerning promoting the health, and prevention of disease [11]. Therefore, all pregnant women are recommended to have their first ANC visit within the first trimester of pregnancy at or before 16 weeks of gestation and to have a minimum of four ANC visits during pregnancy [11]. ANC is an important opportunity to identify and manage any medical complications [12]. Globally, 86% of pregnant women access ANC with skilled health personnel at least once and 65%

receive at least four ANC visits. Whereas, in sub-Saharan Africa, only 52% and South Asia only 49% of women and had received at least four ANC visits [13]. In Ethiopia, in 2016 only 62% of women had any ANC visits during pregnancy, 32% of women had at least four ANC visits, and only 20% of women had their first ANC visit during the first trimester [9].

The World Health Organization (WHO) recommends that mothers and newborns should receive PNC in health facilities within at least 24 hours after birth if birth occurs in a health facility. For home births, the first postnatal contact should be as early as possible within 24 hours of birth. Additionally, at least three postnatal visits, on day 3 (48–72 hours), between 7–14 days, and at six weeks after birth are recommended for all mothers and newborns [14]. Safe motherhood programmers recommend that all women receive a health check within 2 days after delivery [9].

For both the mother and infant, early postnatal care is vital for providing essential information as well as detecting and treating childbirth-related complications. Most maternal and neonatal deaths occur during or immediately after childbirth [1, 9, 14]. Around half of the maternal and newborn deaths occur in the first 24 hours [15, 16]. The majority of maternal deaths can be reduced through increasing antenatal care, skilled care during childbirth, emergency obstetric care and postnatal care services [17, 18]. In Ethiopia, receiving appropriate postnatal care in the recommended time could avert neonatal mortality by 10–27% [16]. However, in spite of its potential role in reducing newborn and maternal deaths, postnatal care has been poorly utilized for the health and survival of both the mother and the newborn [14, 16]. For instance in Ethiopia, only 17% of women and 13% of newborns received their postnatal checkup within the first two days of delivery [9].

Improving the quality of antenatal care services is likely to contribute to rapid increases in postnatal care utilization and results in better health outcomes for women and newborns [19]. As a result, ANC utilization and timely initiation of PNC are considered critical to maternal health services to improve health outcomes for women and newborns. The objective of this study was to determine the association between antenatal care and timely postnatal care checkup among reproductive-age women in Ethiopia. We hypothesized that Ethiopian women who had timely and at least four ANC visits would be more likely to have a timely PNC checkup.

## Methods

### Study setting, design and period

This study was a secondary data analysis of the 2016 Ethiopia Demographic and Health Survey (EDHS) which was collected by the Central Statistical Agency (CSA), Ethiopia and the DHS Program, ICF [9]. The survey was conducted from January 18 to June 27, 2016. A detailed description of the 2016 EDHS study design and methods are available elsewhere [9]. In brief, the 2016 EDHS participants included a stratified, two-stage cluster probability sample representative of the Ethiopian population. The sampling frame for the 2016 EDHS was the 2007 Ethiopian Population and Housing Census (PHC). In the first stage, a total of 645 enumeration areas (EAs) (202 from urban areas and 443 from rural areas) were selected out of the 84,915 EAs listed in the PHC based on probability proportional to EA size. This was followed by the selection of 28 households from each EA from a new list of households prepared for all the selected EAs with an equal probability systematic selection. All women aged 15–49 and men aged 15–59 who were either permanent residents of the selected households or visitors, or who stayed in the household the night before the survey were eligible to be interviewed.

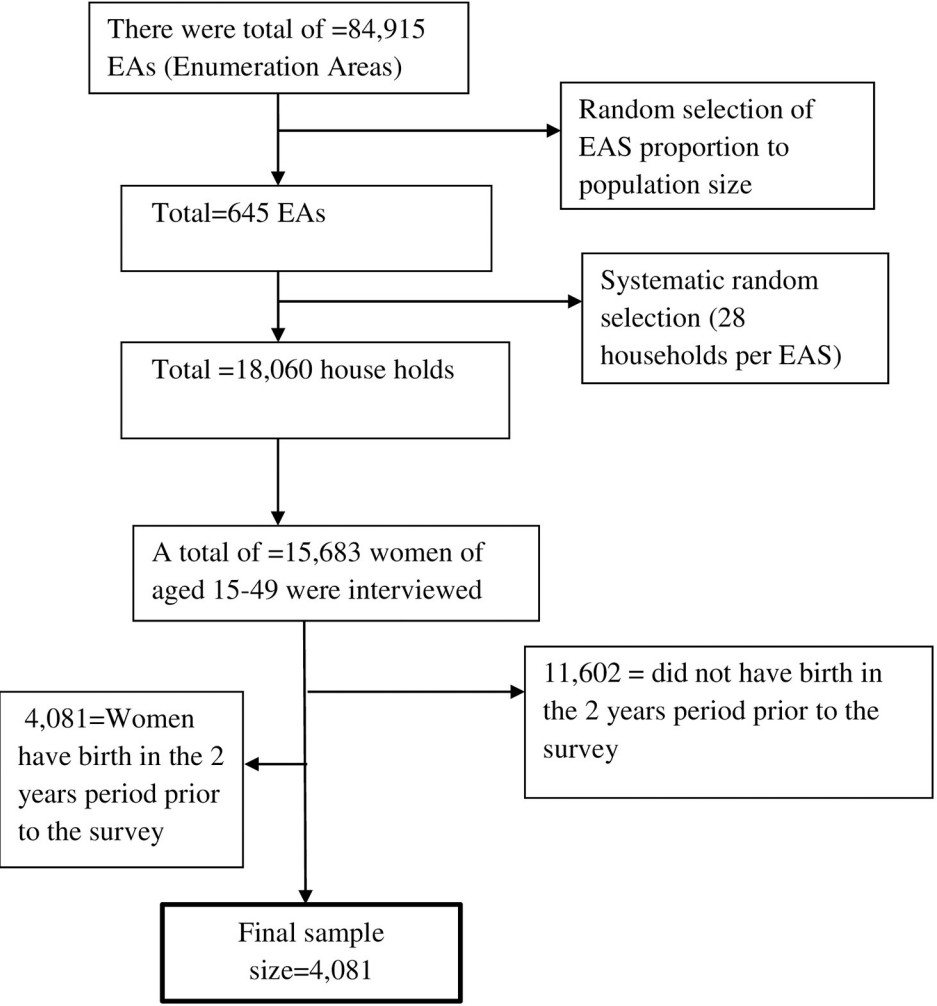

**Fig 1. Schematic presentation of sample selection from EDHS 2016 data set.**

## Inclusion and exclusion criteria

Of the total 15,683 women interviewed in the 2016 EDHS, only those who give birth in the two years preceding the survey were included in this analysis (n = 4,308). Women who had missing data on the variables of interest were excluded from the analysis (n = 227). The final sample consisted of 4,081 participants (Fig 1).

## Study variables

The outcome variable was a timely postnatal checkup, which was defined as receiving a postnatal checkup within 2 days after delivery [9]. The main exposure of interest was antenatal care (ANC) utilization. This was defined based on the WHO recommendations regarding the timing of first ANC attendance and the total number of ANC attendance. As WHO recommends the first ANC visit should take place within the first trimester of gestation, and at least four visits during the course of the pregnancy [20]. According to these guidelines, the exposure variable antenatal care use was categorized as: No ANC visit, 1–3 ANC visits but not started

timely, 1–3 ANC visits and started timely, ≥4 ANC visits and not started timely, ≥4 ANC visits and started timely.

Additional covariates including socio-demographic and obstetric characteristics were selected based on previous research [21–25]. Socio-demographic characteristics included maternal age at the time of birth of the most recent child, highest educational level, marital status, place of residence, current working status of the mother, and household wealth index. DHS uses the principal component analysis technique to assign households wealth scores based on the number and kinds of consumer goods they own, ranging from a television to a bicycle or car, in addition to housing characteristics such as source of drinking water, toilet facilities, and flooring materials [9]. Obstetric characteristics included type of birth attendant (skilled, unskilled), place of delivery (institutional, home), parity [primiparous (1), multiparous (2–4), and grand multiparous (≥5)], mother's perceived distance of the nearest health facility (problematic, non-problematic) and type of pregnancy (planned, unplanned).

## Statistical analysis

Frequencies and percentages were used to describe the socio-demographic characteristics of participants. Chi-square tests were performed to check for factors that were associated with timely postnatal care utilization. Multiple logistic regression analysis was performed to identify factors associated with timely postnatal care checkups. All independent variables were fitted into the multivariable logistic regression to control for confounding and identify the association between antenatal care and timely postnatal care. Adjusted Odds Ratio (aOR) with 95% Confidence Interval (95%CI) was reported. Multicollinearity was checked using variance inflation factor (VIF) with maximum threshold 4 and no multicollinearity was detected. Survey design elements including stratification, cluster and sampling weights were applied to account for complex survey design and unequal probabilities of selection. All analyses were performed using SPSS version 23 [26]. P-value <0.05 were considered statistically significant.

## Ethics approval and consent to participate

Publicity available data set was obtained from the DHS website (https://dhsprogram.com/) through registering with the DHS website and as such, no ethical approval was required. This study was deemed exempt by the Ohio University Institutional Review Board.

## Results

### Sample characteristics

The majority (87.7%) of women lived in rural areas. Furthermore, 95.1%, 60.4% and 45.2% of women were married/living with a partner, had no formal education and lived in the poor wealth quintile, respectively. About 60.5% of women perceived that the distance to a nearby health facility as a barrier to PNC utilization. About 42.2% of the women were multiparous (had two to four pregnancies) and 37.4% were grand multiparous (had five or more pregnancies). Only 36.9% of women gave the most recent birth in the health facility while 37.7% were served by skilled birth attendants. Regarding ANC utilization patterns, 35.7% of women did not have any ANC visits. Only one-fourth (25.1%) of women had timely initiated recommended number of ANC visits (Table 1).

### Timely postnatal checkup

Among women who gave birth in the two years preceding the 2016 EDHS, only 16.5% of women had timely initiated postnatal checkups within two days of delivery. Furthermore,

**Table 1. Descriptive statistics of women who gave birth in the 2 years before the survey, Ethiopia, 2016 DHS (N = 4081).**

| Characteristics | n(weighted%) |
|---|---|
| **Mother's age** | |
| 15–19 | 275 (6.4) |
| 20–34 | 2990 (73.6) |
| ≥35 | 755 (20.0) |
| **Married/living with partner** | |
| No | 198 (4.9) |
| Yes | 3822 (95.1) |
| **Highest educational level** | |
| No formal education | 2383 (60.4) |
| Primary | 1115 (30.6) |
| Secondary or above | 522 (9.0) |
| **Place of residence** | |
| Urban | 829 (12.3) |
| Rural | 3191 (87.7) |
| **Employed** | |
| No | 3005 (75.0) |
| Yes | 1015 (25.0) |
| **Wealth index** | |
| Poor | 2081 (45.2) |
| Middle | 553 (20.6) |
| Rich | 1386 (34.2) |
| **Perceived distance barrier the nearby health facility** | |
| Not significant barrier | 1849 (39.5) |
| Significant barrier | 2171 (60.5) |
| **Type of pregnancy** | |
| Unwanted | 220 (8.5) |
| Wanted | 3800 (91.5) |
| **Parity/birth order** | |
| Primiparous/1 | 870 (20.3) |
| Multiparous/2-4 | 1758 (42.2) |
| Grand multiparous/5+ | 1392 (37.4) |
| **Place of delivery** | |
| Home | 2348 (63.1) |
| Health institution | 1672 (36.9) |
| **Type of birth attendant** | |
| Unskilled | 2302 (62.3) |
| Skilled | 1718 (37.7) |
| **ANC follow-up status and time of initiation** | |
| No | 1332 (35.7) |
| 1–3 not started timely | 729 (20.6) |
| 1–3 started timely | 485 (10.6) |
| ≥4 not started timely | 315 (8.0) |
| ≥4 started timely | 1156 (25.1) |

postnatal checkup significantly differed by a history of ANC follow-up, educational status, place of residence, working status, perceived distance from a nearby health facility, household wealth index, place of delivery and type of birth attendant (Table 2).

## Association between antenatal care utilization pattern and timely postnatal checkup

After controlling for all other variables, antenatal care use was significantly associated with timely postnatal care checkups. Compared to women with no ANC visit, women who had at least four timely ANC visits had higher odds of timely postnatal checkups (aOR: 2.50; 95%CI: 1.42–4.42). Similarly, compared to women with no ANC visit, women who had at least four delayed ANC visits had higher odds of timely postnatal checkups (aOR: 2.46; 95%CI: 1.22–4.97). Additionally, women who had secondary and above education (aOR: 1.64; 95%CI: 1.03–2.60) compared to those who had no formal education, those who perceived distance to the nearby health facility was not a significant barrier (aOR: 1.55; 95%CI: 1.15–2.09) compared to those who perceived distance as a significant problem, primiparous (aOR: 0.34; 95%CI: 0.19–0.61) compared to grand multiparous, and those who had institutional delivery (aOR: 14.55; 95%CI: 2.21–95.77) compared to those who had home delivery were also significantly associated with timely initiation of PNC checkup (Table 3).

## Discussion

In a nationally representative sample of Ethiopian women, we found that having the recommended ANC visits were positively associated with timely utilization of PNC services. This association appeared to be independent of confounders such as maternal socio-demographic and obstetric characteristics. Our results contribute to the emerging evidence on the importance of having the recommended number of ANC visits to the timely initiation of PNC checkups.

Women who started their first antenatal visit in the recommended timeframe and had at least four antenatal care visits were more likely to utilize PNC than their counterparts. This finding is consistent with the findings of several studies that showed that a higher number of ANC visits is an important determinant of PNC utilization [25, 27–32]. Because ANC is a time when mothers are educated about childbirth and childcare practices, those who follow the ANC guidelines might be better informed about PNC guidelines. This assertion is supported through qualitative studies. Belachew et al. (2016) illustrated this point with the following focus group discussion quote, "I had repeated ANC visits during my previous two pregnancies and the health providers provided me cares such as complete physical assessment, I was given red tablets for free and told to swallow one per day, I was counseled about danger signs of pregnancy and also I was told the benefits of institutional delivery and postnatal care. That is why I prefer to deliver my children in a health facility and I attended postnatal care.. . ."page 6, [30]. Increased numbers of antenatal visits might give opportunities to women to learn about birth preparedness, potential complications during pregnancy and labor [33]. The provision of counseling and health education to mothers by skilled health care providers at the time of ANC visits also plays a critical role. As a result, they might be more likely to follow recommendations, including recommended PNC protocol [24].

Compared to women gave birth at home, women who gave birth at health institution had higher odds of having timely PNC checkups compared to those women gave birth at home. This finding was consistent with other studies that showed that women who gave birth at health institutions had higher odds of having PNC checkups [25, 27, 28, 30–32, 34, 35]. Since all women who deliver at health facilities with an uncomplicated vaginal birth are expected to

**Table 2. Characteristics of the study sample by postnatal checkup within 2 days of delivery (N = 4081).**

| Characteristics | Timely postnatal check-up | | p-Value |
|---|---|---|---|
| | **Yes** | **No** | |
| | **n (Weighted %)** | **n (Weighted %)** | |
| **Mother's age** | | | 0.288 |
| 15–19 | 62 (1.0) | 213 (5.4) | |
| 20–34 | 628 (12.7) | 2362 (60.9) | |
| ≥35 | 135 (2.8) | 620 (17.1) | |
| **Married/living with partner** | | | 0.769 |
| No | 50 (0.9) | 148 (4.0) | |
| Yes | 775 (15.7) | 3047 (79.5) | |
| **Highest educational level** | | | <0.001 |
| No formal education | 296 (6.3) | 2087 (54.1) | |
| Primary | 299 (6.3) | 816 (24.2) | |
| Secondary or above | 230 (3.8) | 292 (5.1) | |
| **Residence** | | | <0.001 |
| Urban | 371(5.5) | 458 (6.7) | |
| Rural | 454 (11.0) | 2737 (76.8) | |
| **Employed** | | | 0.001 |
| No | 553 (11.1) | 2452 (64.0) | |
| Yes | 272 (5.4) | 743 (19.5) | |
| **Wealth index** | | | <0.001 |
| Poor | 213 (4.1) | 1868 (41.2) | |
| Middle | 100 (2.8) | 453 (17.7) | |
| Rich | 512 (9.6) | 874 (24.6) | |
| **Perceived distance barrier to the nearby health facility** | | | <0.001 |
| Not significant barrier | 567 (10.5) | 1282 (29.0) | |
| Significant barrier | 258 (6.0) | 1913 (54.5) | |
| **Type of pregnancy** | | | 0.445 |
| Unwanted | 41 (1.2) | 179 (7.3) | |
| Wanted | 784 (15.3) | 3016 (76.2) | |
| **Parity/birth order** | | | <0.001 |
| Primiparous/1 | 255 (4.2) | 615 (16.2) | |
| Multiparous/2-4 | 376 (8.0) | 1382 (34.3) | |
| Grand multiparous/5+ | 194 (4.4) | 1198 (33.1) | |
| **Place of delivery** | | | <0.001 |
| Home | 41 (0.9) | 2307 (62.1) | |
| Health institution | 784 (15.6) | 888 (21.3) | |
| **Type of birth attendant** | | | <0.001 |
| Unskilled | 37 (0.9) | 2265 (61.3) | |
| Skilled | 788 (15.6) | 930 (22.1) | |
| **ANC follow up status and time of initiation** | | | <0.001 |
| No | 46 (1.3) | 1286 (34.4) | |
| 1–3 not started timely | 129 (2.8) | 600 (17.8) | |
| 1–3 started timely | 116 (2.0) | 369 (8.6) | |
| ≥4 not started timely | 80 (2.2) | 235 (5.8) | |
| ≥4 started timely | 454 (8.2) | 705 (16.9) | |

**Table 3. Association between ANC follow up pattern and timely postnatal checkup (N = 4081).**

| Variables | Unadjusted OR* (95% CI) | p-Value | Adjusted OR (95% CI) | p-Value |
|---|---|---|---|---|
| **Mother's age** | | | | |
| 15–19 | 1.08 (0.66,1.77) | 0.757 | 1.66 (0.80, 3.46) | 0.172 |
| 20–34 | 1.28 (0.94,1.68) | 0.127 | 1.20 (0.73, 1.96) | 0.477 |
| ≥35 | **1.00** | | **1.00** | |
| **Married/living with partner** | | | | |
| No | **1.00** | | **1.00** | |
| Yes | 0.91 (0.74, 1.54) | 0.726 | 0.89 (0.52, 1.51) | 0.657 |
| **Highest educational level** | | | | |
| No formal education | **1.00** | | **1.00** | |
| Primary | 2.21 (1.67, 2.92) | <0.001 | 1.33 (0.93, 1.90) | 0.119 |
| Secondary and above | 6.28 (4.43, 8.89) | <0.001 | 1.64(1.03, 2.60) | 0.036 |
| **Place of residence** | | | | |
| Urban | 5.72 (4.21, 7.76) | <0.001 | 1.15 (0.76, 1.74) | 0.508 |
| Rural | **1.00** | | **1.00** | |
| **Employed** | | | | |
| No | **1.00** | | **1.00** | |
| Yes | 1.63 (1.24, 2.14) | <0.001 | 1.01 (0.74, 1.37) | 0.963 |
| **Wealth Index** | | | | |
| Poor | **1.00** | | **1.00** | |
| Middle | 1.69 (1.16, 2.47) | <0.006 | 1.07 (0.66, 1.74) | 0.779 |
| Rich | 3.92 (2.91, 5.28) | <0.001 | 1.03 (0.69, 1.52) | 0.897 |
| **Perceived distance barrier to the nearby health facility** | | | | |
| Not significant barrier | 3.24 (2.54, 4.13) | <0.001 | 1.55 (1.15, 2.09) | 0.004 |
| Significant barrier | **1.00** | | **1.00** | |
| **Type of pregnancy** | | | | |
| Unwanted | **1.00** | | **1.00** | |
| Wanted | 1.23 (0.77, 1.95) | 0.387 | 0.78 (0.43, 1.48) | 0.469 |
| **Parity/birth order** | | | | |
| Primiparous/1 | 1.94 (1.38, 2.71) | <0.001 | 0.34 (0.19, 0.61) | <0.001 |
| Multiparous/ 2–4 | 1.77 (1.36, 2.31) | <0.001 | 0.73 (0.48,1.12) | 0.153 |
| Grand multiparous/ 5+ | **1.00** | | **1.00** | |
| **Place of delivery** | | | | |
| Home | **1.00** | | **1.00** | |
| Health institution | 49.23 (28.83,84.14) | <0.001 | 14.55 (2.21,95.77) | 0.005 |
| **Type of birth attendant** | | | | |
| Unskilled | **1.00** | | **1.00** | |
| Skilled | 38.75 (23.52, 63.85) | <0.001 | 2.54 (0.37,17.56) | 0.344 |
| **ANC follow up status and time of initiation** | | | | |
| No | **1.00** | | **1.00** | |
| 1–3 not started timely | 3.92 (2.34, 6.57) | <0.001 | 1.48 (0.82, 2.66) | 0.197 |
| 1–3 started timely | 5.54 (3.16, 9.70) | <0.001 | 1.74 (0.86, 3.53) | 0.124 |
| ≥4 not started timely | 8.55 (4.68,15.57) | <0.001 | 2.46 (1.22, 4.97) | 0.012 |
| ≥4 started timely | 12.04 (7.38, 19.62) | <0.001 | 2.50 (1.42, 4.42) | 0.002 |

*OR: Odds Ratio;

‡CI: Confidence Interval.

stay in the facility for at least 24 hours after birth, they are likely to receive PNC before discharge [14]. Additionally, women may be advised to return for future visits and be more comfortable with the setting than those who deliver at home. Women who deliver at a facility are also more likely to have the ability to overcome cultural, geographic, financial and other barriers related to health care access [27].

Primiparous women were 66% less likely to utilize postnatal checkups timely as compared with grand multiparous women. Whereas past researchers have found no association between parity and PNC utilization [35–37], the present study has shown a lower likelihood of timely PNC checkups among primiparous women. Previous experience with pregnancy complications may increase the likelihood of seeking post-natal care within two days to avoid similar complications. The fear associated with previous complications may compel women to seek regular care [35]. Primiparous women, with no history to learn from, are, therefore, less likely to have a fear of complications [29]. More research is needed to confirm this finding. In addition, this analysis found that the odds of having a timely PNC checkup were higher among women whose educational status was secondary/higher as compared to those with no formal education. This result was in line with a number of studies showing a higher likelihood of PNC checkups among women with some education [25, 27–29]. The higher maternal educational attainment, the more likely the awareness and knowledge of benefits to be derived to comply with more health recommendations; this is in part due to the fact that they may understand better. Additionally, women who are educated are more likely to have paid employment and to contribute to the household expenditure and consumption and that means more power in the decision-making process in household issues including utilization of health services [32].

Mothers who did not perceive the distance to the nearest health facility as a problem had higher odds of timely PNC checkups as compared to those who perceived distance to a health facility as a significant barrier. This finding is consistent previous studies [27, 29, 38]. Physical proximity to health services is a major problem, especially in rural villages with poor road conditions. Participants in a qualitative study complained that they needed to walk for up to two hours to reach the nearest health center. The situation became worse during the rainy season when the road was slippery [25]. It is, therefore, not surprising that distance from a healthcare facility is a significant barrier to timely and recommended PNC utilization.

## Strength and limitations of the study

The main strengths of the study are the use of nationally representative survey data and the availability of several potential confounders for adjustment in the multivariable regression model. Additionally, data were collected using a standardized questionnaire with rigorous procedures to check for data quality. However, this study has several limitations. As secondary data analysis, some important variables were not included, such as the role of husbands and other family members in maternal health decision-making and cultural beliefs about when women are allowed to leave the house in the postpartum period. Additionally, since women were asked retrospectively for their exposures, the response might be susceptible to recall bias. Furthermore, the cross-sectional nature of the study does not allow for the determination of causation.

## Conclusions

The initiation of postnatal care within two days of delivery in Ethiopia is very low. Utilizing the recommended number of ANC visits, starting it at an early stage of pregnancy was positively associated with the timely initiation of postnatal care. Even women, who had not initiated ANC in the recommended time frame but had had the recommended number of visits,

had higher odds of appropriate PNC. Thus, strengthening the existing maternal and child health program to adhere to the recommended number of ANC is likely to improve adherence to PNC protocols.

## Acknowledgments

We are grateful to the DHS Program for providing us permission to use the 2016 EDHS data for this analysis.

## Author Contributions

**Conceptualization:** Gizachew Tadesse Wassie, Minyichil Birhanu Belete, Azimeraw Arega Tesfu, Simachew Animen Bantie, Asteray Assmie Ayenew, Belaynew Adugna Endeshaw, Semaw Minale Agdie, Mengistu Desalegn Kiros, Zelalem T. Haile, Mohammad Rifat Haider, Gillian H. Ice.

**Data curation:** Minyichil Birhanu Belete, Simachew Animen Bantie, Belaynew Adugna Endeshaw, Semaw Minale Agdie, Mengistu Desalegn Kiros, Zelalem T. Haile.

**Formal analysis:** Minyichil Birhanu Belete, Simachew Animen Bantie, Asteray Assmie Ayenew, Belaynew Adugna Endeshaw, Gillian H. Ice.

**Methodology:** Gizachew Tadesse Wassie, Minyichil Birhanu Belete, Simachew Animen Bantie, Mohammad Rifat Haider, Gillian H. Ice.

**Software:** Zelalem T. Haile, Mohammad Rifat Haider.

**Supervision:** Zelalem T. Haile, Mohammad Rifat Haider, Gillian H. Ice.

**Writing – original draft:** Gizachew Tadesse Wassie, Azimeraw Arega Tesfu, Asteray Assmie Ayenew, Semaw Minale Agdie.

**Writing – review & editing:** Gizachew Tadesse Wassie, Minyichil Birhanu Belete, Azimeraw Arega Tesfu, Zelalem T. Haile, Mohammad Rifat Haider, Gillian H. Ice.

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
