## [Decision Letter · Decision Letter 0]

9 Feb 2021

PONE-D-20-37280

Association between antenatal care utilization pattern and timely initiation of postnatal care checkup: Analysis of 2016 Ethiopian Demographic and Health Survey

PLOS ONE

Dear Dr. Gizachew Tadesse Wassie,

Thank you for submitting your manuscript to PLOS ONE. After careful consideration, we feel that it has merit but does not fully meet PLOS ONE’s publication criteria as it currently stands. Therefore, we invite you to submit a revised version of the manuscript that addresses the points raised during the review process.

Refer to the comments below.

We look forward to receiving your revised manuscript.

Kind regards,

Mohd Noor Norhayati, M.B.B.S., M.Comm.Med., Ph.D.

Academic Editor

PLOS ONE

Journal Requirements:

Additional Editor Comments:

To relook into the linkage between the use of antenatal care and postnatal care. And among those with postnatal care, to differentiate between home and institutional deliveries. And the connection with these two settings for childbirth.

Reviewers' comments:

Reviewer's Responses to Questions

**Comments to the Author**

1. Is the manuscript technically sound, and do the data support the conclusions?

Reviewer #1: Partly

2. Has the statistical analysis been performed appropriately and rigorously? 

Reviewer #1: Yes

3. Have the authors made all data underlying the findings in their manuscript fully available?

Reviewer #1: Yes

4. Is the manuscript presented in an intelligible fashion and written in standard English?

Reviewer #1: Yes

5. Review Comments to the Author

Reviewer #1: General comments:

The authors have highlighted a critical issue in Ethiopia: low prevalence of timely use of postnatal care. They hypothesize an association between antenatal care and postnatal care and use a nationally representative dataset to test that association. However, the paper is missing a critical analysis and discussion of the expected pathway/linkages between use of antenatal care and postnatal care; this re-analysis and re-write would be necessary in order to make useful recommendations for future study or intervention.

Major comments (should be addressed before publication)

1. The manuscript would benefit from a detailed description of what the authors think the theory of effect is here. How would ANC affect PNC? Are they both driven by a confounder that affects access? Is the experience of ANC (e.g. quality of care) driving the decision to go for PNC? Are current covariates included because they are confounders or just trying to soak up variation? A description and a DAG would be helpful for understanding the importance of this analysis, what it might mean, and for the analytic choices made.

a. Note, assessing counseling as a predictor of PNC could provide more strength to the hypothesis shared in the discussion section :“Because ANC is a time when mothers are educated about childbirth and childcare practices, those who follow the ANC guidelines might be better informed about PNC guidelines.”

b. Also in the discussion, the following assertion can be tested using DHS data and therefore should be tested if you want to discuss it in the discussion: “Increased numbers of antenatal visit might give opportunities to women to learn about birth preparedness, potential complications during pregnancy and labor”

2. Given the very large difference between those who delivered in an institution versus home, I would recommend stratifying those populations. Who got a postnatal visit among those who delivered in an institution, and who among those who delivered at home? This ties again into the theory of change – does ANC lead more women to deliver in an institution, which then leads them to have a timely PNC visit, or does ANC lead to increased PNC regardless of institutional delivery? Note that this phrasing is causal, because I’m thinking about theory, but this analysis will only look at associations not causation.

3. Again, depending on your theory of change, it is likely that one pathway through which ANC could affect PNC use is through quality of care (the authors allude to this in the last paragraph of the introduction). The analysis would be strengthened if this hypothesis were tested as well. The authors could create a quality of care index based on the services reported as received during ANC (while patient report is not the gold standard for these process of care measures, it might provide some indication of quality of care).

4. Discussion: please discuss how 36.9% of women gave birth in the health facility, but only 16.5% had a postnatal check-up within two days. Are women being discharged without a postnatal check-up? How long after delivery does the check up have to occur to be considered “postnatal”? This second question ties into the confusion above of discussing hemorrhage and birth asphyxia in relation to postnatal care. It would be helpful to see the question from the DHS that this analysis is based on (does it mean that ~half o the people who delivered in the facility answered “no” to the question “Did anyone check on your health while you were still in the facility?” The authors discuss that women are supposed to stay for 24 hours, so this issue of women who deliver in a health facility not receiving a PNC visit seems to be one of quality of care, not of the woman’s choice or access.

5. The paper focuses entirely on the first postnatal visit – is it possible to also look at subsequent visits?

6. Together with the careful theory of change discussion requested above, I think the paper would benefit from a careful read to make sure there is no causal language implied. Without understanding the pathways through which ANC and PNC are associated, it’s hard to know if increasing ANC would meaningfully impact PNC (the authors do include this in their limitations section, which is great).

Minor comments (to improve clarity)

1. Introduction: the first and second sentence almost seem at odds with one another. Hemorrhage and birth asphyxia within the first day would be most affected by delivery at a health facility. Maybe adding other causes of morbidity and mortality that would also be affected by the subsequent postnatal visits would be helpful.

2. Figure 1: It’s helpful to break out the women who were not eligible (didn’t deliver in past two years) from those who had

3. Table 1: Is education the highest level completed? Please clarify in the table

4. Sample characteristics: The sentence “Nearly half (42.2%) of the women were multiparous (had two or more pregnancy).” is misleading. I suggest reporting the number of primiparous women (preferable), or report the full number of multiparous, which according to the table would be 42.2%+37.4%

5. Typo in the aOR for secondary education, missing the decimal place (164)

6. Discussion: this sentence seems to confuse multiparous with grand multiparous, which appears to be the comparison group in table 1 “Primiparous women were 66 % less likely to utilize postnatal checkup timely as compared with multiparous women.”

6. PLOS authors have the option to publish the peer review history of their article (what does this mean?). If published, this will include your full peer review and any attached files.

Reviewer #1: No

---

## [Author Response · Author response to Decision Letter 0]

24 Mar 2021

Response to the Reviewers

Dear editor, thank you for considering our manuscript for publication; we have revised the contents based on the reviewer’s comments as below.

Ref: Submission ID : PONE-D-20-37280

"Association between antenatal care utilization pattern and timely initiation of postnatal care checkup: Analysis of 2016 Ethiopian Demographic and Health Survey

PLOS ONE."

Additional Editor Comments:

To relook into the linkage between the use of antenatal care and postnatal care. And among those with postnatal care, to differentiate between home and institutional deliveries. And the connection with these two settings for childbirth.

Response: We have adjusted for place of delivery in the analysis.

Reviewers' comments:

Reviewer's Responses to Questions

Comments to the Author

1. Is the manuscript technically sound, and do the data support the conclusions?

Reviewer #1: Partly

2. Has the statistical analysis been performed appropriately and rigorously?

Reviewer #1: Yes

3. Have the authors made all data underlying the findings in their manuscript fully available?

Reviewer #1: Yes

4. Is the manuscript presented in an intelligible fashion and written in standard English?

Reviewer #1: Yes

5. Review Comments to the Author

Reviewer #1: General comments:

The authors have highlighted a critical issue in Ethiopia: low prevalence of timely use of postnatal care. They hypothesize an association between antenatal care and postnatal care and use a nationally representative dataset to test that association. However, the paper is missing a critical analysis and discussion of the expected pathway/linkages between use of antenatal care and postnatal care; this re-analysis and re-write would be necessary in order to make useful recommendations for future study or intervention.

Response: Thank you.

Major comments (should be addressed before publication)

1. The manuscript would benefit from a detailed description of what the authors think the theory of effect is here. How would ANC affect PNC? Are they both driven by a confounder that affects access? Is the experience of ANC (e.g. quality of care) driving the decision to go for PNC? Are current covariates included because they are confounders or just trying to soak up variation? A description and a DAG would be helpful for understanding the importance of this analysis, what it might mean, and for the analytic choices made.

a. Note, assessing counseling as a predictor of PNC could provide more strength to the hypothesis shared in the discussion section :“Because ANC is a time when mothers are educated about childbirth and childcare practices, those who follow the ANC guidelines might be better informed about PNC guidelines.”

b. Also in the discussion, the following assertion can be tested using DHS data and therefore should be tested if you want to discuss it in the discussion: “Increased numbers of antenatal visit might give opportunities to women to learn about birth preparedness, potential complications during pregnancy and labor”

Response: We appreciate this feedback. 

a. As indicated in the introduction section we tested the hypothesis that women who had timely and at least four ANC visits would be more likely to have a timely PNC checkup. Covariates were selected based on existing literature on factors influencing timely PNC checkup. We have now indicated this in the methods section (Page 6, Line 151-152). As the reviewer indicated precisely, we were testing if antenatal counselling is associated with timely PNC checkup. 

b. This was one possible explanation for the observed relationship between number of antenatal care visit and increased likelihood of PNC (Page 13, Line 227-230).

2. Given the very large difference between those who delivered in an institution versus home, I would recommend stratifying those populations. Who got a postnatal visit among those who delivered in an institution, and who among those who delivered at home? This ties again into the theory of change – does ANC lead more women to deliver in an institution, which then leads them to have a timely PNC visit, or does ANC lead to increased PNC regardless of institutional delivery? Note that this phrasing is causal, because I’m thinking about theory, but this analysis will only look at associations not causation.

Responses: Thank you for the feedback. As you stated clearly this study examined the association between ANC number and timing with timely receipt of PNC. By controlling place of delivery in the multivariable regression model we are taking these differences into account. We were not interested in effect measure modification by place of delivery, which of course can be a separate manuscript by itself.

3. Again, depending on your theory of change, it is likely that one pathway through which ANC could affect PNC use is through quality of care (the authors allude to this in the last paragraph of the introduction). The analysis would be strengthened if this hypothesis were tested as well. The authors could create a quality of care index based on the services reported as received during ANC (while patient report is not the gold standard for these process of care measures, it might provide some indication of quality of care).

Response: Thank you for this feedback. The focus of our research was the timing and number of ANC visits as recommended by the World Health Organization (WHO), i.e., least four ANC visits during pregnancy and first ANC during the first trimester. Assessing the components on ANC was out of scope for this manuscript. 

4. Discussion: please discuss how 36.9% of women gave birth in the health facility, but only 16.5% had a postnatal check-up within two days. 

Are women being discharged without a postnatal check-up? 

Response: Ideally, postnatal care is best delivered in a health facility. However, in Ethiopian context due to many socio-economic and cultural reasons, such as the lack of MNH service, lack of trained professional, lack of bed and essential medicines women are obligated to be discharged early without a postnatal checkup to get space for the complicated ones. Therefore, it is likely that some women are discharged without postnatal check-up.

5. How long after delivery does the checkup have to occur to be considered “postnatal”? 

Response: Postnatal care (PNC) is the care given to the mother and her newborn baby immediately after the birth and for the first six weeks of life. In the current study we used WHO’s 2018 Global reference list of 100 core health indicators (‎plus health-related SDGs)‎ to define the timely PNC as proportion of women who have postpartum contact with a health provider within 2 days of delivery (Page 5, Line 142-143).

6. This second question ties into the confusion above of discussing hemorrhage and birth asphyxia in relation to postnatal care. It would be helpful to see the question from the DHS that this analysis is based on (does it mean that ~half of the people who delivered in the facility answered “no” to the question “Did anyone check on your health while you were still in the facility?” The authors discuss that women are supposed to stay for 24 hours, so this issue of women who deliver in a health facility not receiving a PNC visit seems to be one of quality of care, not of the woman’s choice or access.

Response: Even though the recommendations given by WHO women are supposed to stay at health facility for 24 hours, practically women discharged on average for six hours. This is mainly due to lack of space. However, women are told to come back for postnatal checkup. This may be due to sub-standard quality of service practiced by health professionals as you mentioned.

5. The paper focuses entirely on the first postnatal visit – is it possible to also look at subsequent visits?

Response: This is possible. However, the focus of this paper was timely initiation of postnatal care; defined with the current WHO indicators as within two days after delivery. We also wanted to focus on this period because around 45-50% of mothers and newborns die in the first 24 hours after birth. So, if PNC is initiated early, many unforeseen complications may be identified and intervene early. 

6. Together with the careful theory of change discussion requested above, I think the paper would benefit from a careful read to make sure there is no causal language implied. Without understanding the pathways through which ANC and PNC are associated, it’s hard to know if increasing ANC would meaningfully impact PNC (the authors do include this in their limitations section, which is great).

Response: Thank you. As we stated in the introduction, we were interested in assessing the hypothesis that receipt of the recommended number of ANC may influence timely PNC. 

Minor comments (to improve clarity)

1. Introduction: the first and second sentence almost seem at odds with one another. Hemorrhage and birth asphyxia within the first day would be most affected by delivery at a health facility. Maybe adding other causes of morbidity and mortality that would also be affected by the subsequent postnatal visits would be helpful.

Response: We have added some other complications /morbidities which occur following delivery.

2. Figure 1: It’s helpful to break out the women who were not eligible (didn’t deliver in past two years) from those who had.

Response: Corrected as recommended (figure 1).

3. Table 1: Is education the highest level completed? Please clarify in the table

Response: Yes, it is the highest educational level. We have changed the label in the tables to reflect this. 

4. Sample characteristics: The sentence “Nearly half (42.2%) of the women were multiparous (had two or more pregnancy).” is misleading. I suggest reporting the number of primiparous women (preferable), or report the full number of multiparous, which according to the table would be 42.2%+37.4%

Response: We have corrected as recommended.

5. Typo in the aOR for secondary education, missing the decimal place (164)

Response: Thank you. We have corrected this to read as corrected as 1.64.

6. Discussion: this sentence seems to confuse multiparous with grand multiparous, which appears to be the comparison group in table 1 “Primiparous women were 66 % less likely to utilize postnatal checkup timely as compared with multiparous women.”

Response: Thank you. We have corrected this as recommended.

---

## [Decision Letter · Decision Letter 1]

24 Jun 2021

PONE-D-20-37280R1

Association between antenatal care utilization pattern and timely initiation of postnatal care checkup: Analysis of 2016 Ethiopian Demographic and Health Survey

PLOS ONE

Dear Dr. Wassie,

Thank you for submitting your manuscript to PLOS ONE. After careful consideration, we feel that it has merit but does not fully meet PLOS ONE’s publication criteria as it currently stands. Therefore, we invite you to submit a revised version of the manuscript that addresses the points raised during the review process.

We look forward to receiving your revised manuscript.

Kind regards,

David Teye Doku

Academic Editor

PLOS ONE

Journal Requirements:

Reviewers' comments:

Reviewer's Responses to Questions

**Comments to the Author**

1. If the authors have adequately addressed your comments raised in a previous round of review and you feel that this manuscript is now acceptable for publication, you may indicate that here to bypass the “Comments to the Author” section, enter your conflict of interest statement in the “Confidential to Editor” section, and submit your "Accept" recommendation.

Reviewer #2: (No Response)

Reviewer #3: All comments have been addressed

2. Is the manuscript technically sound, and do the data support the conclusions?

Reviewer #2: Yes

Reviewer #3: Yes

3. Has the statistical analysis been performed appropriately and rigorously? 

Reviewer #2: Yes

Reviewer #3: Yes

4. Have the authors made all data underlying the findings in their manuscript fully available?

Reviewer #2: Yes

Reviewer #3: Yes

5. Is the manuscript presented in an intelligible fashion and written in standard English?

Reviewer #2: Yes

Reviewer #3: Yes

6. Review Comments to the Author

Reviewer #2: The authors have effectively addressed the majority of the reviewer comments. However, a few important concerns remain unaddressed. These include the following:

1. An important comment by the reviewer "A description and a DAG would be

helpful for understanding the importance of this analysis, what it might mean, and for

the analytic choices made" was not addressed.

2. On the sample characteristics, the comment "Sample characteristics: The sentence “Nearly half (42.2%) of the women were

multiparous (had two or more pregnancy) ” is misleading" was not addressed as claimed by the authors. Here, I believe the reviewer wanted authors to directly state 42.2% as this figure cannot be considered as "nearly half".

3. The "Highest educational level" in Table 1 should also be replicated in Tables 2 and 3 as well as under the "Study variable".

4. This is very important. Some of the journal format guidelines were not adhered to by the authors. For example,

the in-text citations are expected to be in square brackets but not round. Ultimately, reference no. 4 "postpartum-complications[1]" was poorly cited and should be checked and fully provided. This also applies to reference numbers 21 (https://doi.org/10.11564/28-3-638) and 22 in the reference list.

Finally, the journal requires authors to provide the "Declarations" (Authors’ contributions, funding, competing interests, Availability of data and materials among others) in the editorial manager system but not in the manuscript as provided by the authors. Authors have to address this to avoid possible delay in the publication of their paper.

Reviewer #3: Authors have carefully addressed almost all the issues raised by reviewers in the first review process. However, the additional comments below should be addressed to enhance the quality of the paper.

General comments:

There are some few grammatical errors in the manuscript including missing commas, missing function words and dots. E.g. (lines; 41, 75, 82-83, 136, 175-175, 217 etc.). Authors should carefully read through the work and correct all grammatical errors in the papers.

The sentence on line 41-42 should be changed to “Chi-square test and multivariable logistic regression analyses were used to...”

The sentence on line 164 should also be changed to “Multiple logistic regression analyses was performed…”

Authors should avoid the use of complex sentences which are difficult to understand e.g. (lines 88-89)

Authors should be clear on the marital status of participants. What does the “Yes” and “No” categories of the marital statuses of participants in the results represent? E.g. Married or Single

The “None” as a category of educational status in the result section should be changed to “No formal education”

The sentence on line 202- 203 should be changed to …women who had secondary and above education (aOR: 1.64; 95%CI: 1.03-2.60) compared to those who had no formal education…”

Authors should briefly discuss how similar or different their study results are in relation to other studies. Thus, what are the evidence in literature of which results in this study is consistent with? (Lines 217-218, lines 231-232, line 239, lines 245-246, line 254

7. PLOS authors have the option to publish the peer review history of their article (what does this mean?). If published, this will include your full peer review and any attached files.

Reviewer #2: No

Reviewer #3: No

---

## [Author Response · Author response to Decision Letter 1]

10 Jul 2021

Response to the Reviewers

Dear editor, thank you for considering our manuscript for publication; we have revised the contents based on the reviewer’s comments as below.

Ref: Submission ID : PONE-D-20-37280

"Association between antenatal care utilization pattern and timely initiation of postnatal care checkup: Analysis of 2016 Ethiopian Demographic and Health Survey

PLOS ONE."

Reviewers' comments:

Reviewer's Responses to Questions

Comments to the Author

1. If the authors have adequately addressed your comments raised in a previous round of review and you feel that this manuscript is now acceptable for publication, you may indicate that here to bypass the “Comments to the Author” section, enter your conflict of interest statement in the “Confidential to Editor” section, and submit your "Accept" recommendation.

Reviewer #2: (No Response)

Reviewer #3: All comments have been addressed

2. Is the manuscript technically sound, and do the data support the conclusions?

Reviewer #2: Yes

Reviewer #3: Yes

3. Has the statistical analysis been performed appropriately and rigorously?

Reviewer #2: Yes

Reviewer #3: Yes

4. Have the authors made all data underlying the findings in their manuscript fully available?

Reviewer #2: Yes

Reviewer #3: Yes

5. Is the manuscript presented in an intelligible fashion and written in standard English?

Reviewer #2: Yes

Reviewer #3: Yes

6. Review Comments to the Author

Reviewer #2: The authors have effectively addressed the majority of the reviewer comments. However, a few important concerns remain unaddressed. These include the following:

1. An important comment by the reviewer "A description and a DAG would be

helpful for understanding the importance of this analysis, what it might mean, and for the analytic choices made" was not addressed.

Response: Thank you for the feedback. We understand that DAGs can be used map all a priori assumptions surrounding a causal question of interest and identify potential confounders. In the current study we used one of the conventional approaches i.e., selecting potential socio-demographic and obstetric confounders based on previous research which have been cited in the study variables section.

2. On the sample characteristics, the comment "Sample characteristics: The sentence “Nearly half (42.2%) of the women were multiparous (had two or more pregnancy) ” is misleading" was not addressed as claimed by the authors. Here, I believe the reviewer wanted authors to directly state 42.2% as this figure cannot be considered as "nearly half".

Response: "nearly half (42.2%) “has been changed to “About 42.2%”

3. The "Highest educational level" in Table 1 should also be replicated in Tables 2 and 3 as well as under the "Study variable".

Response: Thank you for the feedback. We have made these corrections in tables 2 and 3.

4. This is very important. Some of the journal format guidelines were not adhered to by the authors. For example,

the in-text citations are expected to be in square brackets but not round. Ultimately, reference no. 4 "postpartum-complications[1]" was poorly cited and should be checked and fully provided. This also applies to reference numbers 21 (https://doi.org/10.11564/28-3-638) and 22 in the reference list.

Finally, the journal requires authors to provide the "Declarations" (Authors’ contributions, funding, competing interests, Availability of data and materials among others) in the editorial manager system but not in the manuscript as provided by the authors. Authors have to address this to avoid possible delay in the publication of their paper.

Response: Thank you for your detail review and feedback. We have checked and recited (reff #4, 15, 21&22). 

We have now used the Vancouver referencing style using square brackets for the in-text citations as indicated in the authors guideline for the journal. .

We have also removed the "Declarations" (Authors’ contributions, funding, competing interests, Availability of data and materials among others) from the manuscript as recommended. 

Reviewer #3: Authors have carefully addressed almost all the issues raised by reviewers in the first review process. However, the additional comments below should be addressed to enhance the quality of the paper.

General comments:

There are some few grammatical errors in the manuscript including missing commas, missing function words and dots. E.g. (lines; 41, 75, 82-83, 136, 175-175, 217 etc.). Authors should carefully read through the work and correct all grammatical errors in the papers.

Response: Thank you for the feedback. We revised and corrected per your suggestions (lines; 41, 75, 82-83, 136, 175-175, 217). 

The sentence on line 41-42 should be changed to “Chi-square test and multivariable logistic regression analyses were used to...”

Response: Thank you. This has been corrected as recommended. .

The sentence on line 164 should also be changed to “Multiple logistic regression analyses was performed…”

Response: Thank you for the feedback. Corrected (line 164).

Authors should avoid the use of complex sentences which are difficult to understand e.g. (lines 88-89)

Response: Rephrased (lines 88-89)

Authors should be clear on the marital status of participants. What does the “Yes” and “No” categories of the marital statuses of participants in the results represent? E.g. Married or Single

The “None” as a category of educational status in the result section should be changed to “No formal education”

The sentence on line 202- 203 should be changed to …women who had secondary and above education (aOR: 1.64; 95%CI: 1.03-2.60) compared to those who had no formal education…”

Reponses: Thank you for the feedback. All the above issues have been addressed as recommended (Tables 1, 2, 3).

Authors should briefly discuss how similar or different their study results are in relation to other studies. Thus, what are the evidence in literature of which results in this study is consistent with? (Lines 217-218, lines 231-232, line 239, lines 245-246, line 254

Response: We appreciate this feedback. Discussions have been added regarding the consistency of the findings with existing literature.

---

## [Editor Report · Decision Letter 2]

10 Aug 2021

PONE-D-20-37280R2

Association between antenatal care utilization pattern and timely initiation of postnatal care checkup: Analysis of 2016 Ethiopian Demographic and Health Survey

PLOS ONE

Dear Dr. Wassie,

Thank you for submitting your manuscript to PLOS ONE. After careful consideration, we feel that it has merit but does not fully meet PLOS ONE’s publication criteria as it currently stands. Therefore, we invite you to submit a revised version of the manuscript that addresses the points raised during the review process.

We look forward to receiving your revised manuscript.

Kind regards,

David Teye Doku

Academic Editor

PLOS ONE

Journal Requirements:

Additional Editor Comments (if provided):

Language checking and thorough editing is recommended.
---

## [Author Response · Author response to Decision Letter 2]

17 Aug 2021

Response to the Reviewers

Dear editor, thank you for considering our manuscript for publication; we have revised the contents based on the reviewer’s comments as below.

Ref: Submission ID : PONE-D-20-37280

"Association between antenatal care utilization pattern and timely initiation of postnatal care checkup: Analysis of 2016 Ethiopian Demographic and Health Survey

PLOS ONE."

Journal Requirements:

Response: We have reviewed our reference list. No retracted papers were identified.

Additional Editor Comments (if provided):

Language checking and thorough editing is recommended.

Response: We have edited the manuscript for language.
---

## [Editor Report · Decision Letter 3]

29 Sep 2021

Association between antenatal care utilization pattern and timely initiation of postnatal care checkup: Analysis of 2016 Ethiopian Demographic and Health Survey

PONE-D-20-37280R3

Dear Dr. Wassie,

We’re pleased to inform you that your manuscript has been judged scientifically suitable for publication and will be formally accepted for publication once it meets all outstanding technical requirements.

Kind regards,

David Teye Doku

Academic Editor

PLOS ONE
---

## [Editor Report · Acceptance letter]

1 Oct 2021

PONE-D-20-37280R3 

Association between antenatal care utilization pattern and timely initiation of postnatal care checkup: Analysis of 2016 Ethiopian Demographic and Health Survey 

Dear Dr. Wassie:

I'm pleased to inform you that your manuscript has been deemed suitable for publication in PLOS ONE. Congratulations! Your manuscript is now with our production department. 

Kind regards, 

on behalf of

Dr. David Teye Doku 

Academic Editor

PLOS ONE